# On the Origin of Neo-Sex Chromosomes in the Neotropical Dragonflies *Rhionaeschna bonariensis* and *R. planaltica* (Aeshnidae, Odonata)

**DOI:** 10.3390/insects13121159

**Published:** 2022-12-15

**Authors:** Liliana M. Mola, Iva Vrbová, Daniela S. Tosto, Magda Zrzavá, František Marec

**Affiliations:** 1Laboratory of Cytogenetics and Evolution, Faculty of Exact and Natural Sciences, University of Buenos Aires, Buenos Aires C1428EGA, Argentina; 2Institute of Ecology, Genetics and Evolution of Buenos Aires, National Council of Scientific and Technical Research, Buenos Aires C1428EGA, Argentina; 3Biology Centre CAS, Institute of Entomology, Branišovská 31, 370 05 České Budějovice, Czech Republic; 4Biology Centre CAS, Institute of Plant Molecular Biology, Branišovská 31, 370 05 České Budějovice, Czech Republic; 5Instituto de Agrobiotecnología y Biología Molecular (IABIMO), Instituto Nacional de Tecnología Agropecuaria (INTA), Consejo Nacional de Investigaciones Científicas y Técnicas (CONICET), Hurlingham, Buenos Aires 1686, Argentina; 6Department of Molecular Biology and Genetics, Faculty of Science, University of South Bohemia, Branišovská 1760, 370 05 České Budějovice, Czech Republic

**Keywords:** dragonflies, holokinetic chromosomes, nucleolar organizer region, ribosomal DNA, structural rearrangements, meiosis, neo-sex chromosome evolution, FISH, GISH

## Abstract

**Simple Summary:**

Dragonflies and damselflies (Odonata) are very interesting insects from a cytogenetic point of view. Their chromosomes do not have a typical centromere and their meiosis process differs in some respects from the canonical meiosis process. Sex in Odonata is usually determined by two X chromosomes in females and only one X chromosome in males (a Y chromosome is not present). In this work, we studied sex chromosome evolution in two dragonfly species of the genus *Rhionaeschna* that have a derived sex chromosome system: neo-XX in females and neo-XY in males. This variation is the result of chromosome rearrangements. In *R. planaltica*, meiotic analysis and fluorescence in situ hybridization with a ribosomal DNA probe revealed that the original X chromosome was inserted into the smallest autosome, giving rise to the neo-X chromosome, while the homologous autosome became a neo-Y chromosome. In contrast, the neo-X chromosome in *R. bonariensis* evolved by a terminal fusion of the original X chromosome with the largest autosome, whose homolog became the neo-Y chromosome. Our results suggest an independent origin of neo-sex chromosomes in these dragonfly species and contribute to our understanding of the distinct mechanisms of sex chromosome evolution.

**Abstract:**

Odonata have holokinetic chromosomes. About 95% of species have an XX/X0 sex chromosome system, with heterogametic males. There are species with neo-XX/neo-XY sex chromosomes resulting from an X chromosome/autosome fusion. The genus *Rhionaeschna* includes 42 species found in the Americas. We analyzed the distribution of the nucleolar organizer region (NOR) using FISH with rDNA probes in *Rhionaeschna bonariensis* (n = 12 + neo-XY), *R. planaltica* (n = 7 + neo-XY), and *Aeshna cyanea* (n = 13 + X0). In *R. bonariensis* and *A. cyanea*, the NOR is located on a large pair of autosomes, which have a secondary constriction in the latter species. In *R. planaltica*, the NOR is located on the ancestral part of the neo-X chromosome. Meiotic analysis and FISH results in *R. planaltica* led to the conclusion that the neo-XY system arose by insertion of the ancestral X chromosome into an autosome. Genomic in situ hybridization, performed for the first time in Odonata, highlighted the entire neo-Y chromosome in meiosis of *R. bonariensis*, suggesting that it consists mainly of repetitive DNA. This feature and the terminal chiasma localization suggest an ancient origin of the neo-XY system. Our study provides new information on the origin and evolution of neo-sex chromosomes in Odonata, including new types of chromosomal rearrangements, NOR transposition, and heterochromatin accumulation.

## 1. Introduction

The insect order Odonata includes more than 6300 species worldwide, and this number may increase to about 10,000 species based on reports published over six decades [1,2]. Nearly 1700 species are known from the Neotropical region [3].

This order is subdivided into three suborders: Zygoptera (i.e., damselflies), the monophyletic Anisoptera (i.e., dragonflies), and Anisozygoptera (represented by the single genus *Epiophlebia* with four species from Asia), although other analyses include the latter two groups in the suborder Epiprocta [4,5].

Studying Odonata species and associated groups is a useful tool for assessing and monitoring aquatic habitats, especially wetlands and small water bodies, due to their high sensitivity to environmental and water quality changes. Therefore, they provide valuable information about both the impact of anthropogenic activities (e.g., global warming) and the implementation of appropriate management measures [6,7].

From a cytogenetic point of view, Odonata are characterized by holokinetic chromosomes [8]. The XX/X0 chromosome system of sex determination is present in about 95% of Odonata species, with males being the heterogametic sex. In addition, there is also the neo-XX/neo-XY sex determination system, which results from the fusion of the X chromosome with an autosome. This system was first described in 1943 by Oksala [9], who performed a detailed analysis of the meiotic behavior of chromosomes in five species of the genus *Aeshna*. Moreover, the multiple sex chromosomes X_1_X _1_X_2_X_2_/X_1_X_2_Y in *Micrathyria ungulata* (Libellulidae) arose from two fusions: first, from the fusion of the X chromosome with an autosome and second, from the fusion of the neo-Y chromosome with another autosome [10,11,12,13,14,15,16,17,18].

In insects with holokinetic chromosomes, multiple sex chromosome systems can also result from fragmentation of sex chromosomes, as has been described for Lepidoptera (WZ/ZZ, female/male) and Heteroptera (XY/XX, male/female) [19,20,21,22,23,24,25]. However, such an origin of multiple sex chromosomes has not yet been reported for Odonata.

Neo-sex chromosome systems are unevenly distributed across the families and genera of the Odonata, with the male sex chromosome bivalent being homomorphic in half of the species and heteromorphic in the other half. Such heteromorphism can be detected either throughout meiosis (e.g., *Gynacantha interioris*, *Aeshna grandis*, *A. coerulea*, *Rhionaeschna planaltica*, and *Micrathyria longifasciata*) or only in diplotene and diakinesis because the sex chromosome bivalent is masked by chromosome condensation in metaphase I (e.g., *Erythrodiplax media*, *Crocothemis servilia mariannae, Anax ephippiger, Rhionaeschna bonariensis*, and *Aeshna juncea*). This variation is due to differences in the size of the X chromosome (which is usually the smallest of the complement) and the autosome with which it was fused [9,10,12,13,26,27,28,29,30].

Another characteristic feature of the Odonata is the presence of a very small pair of chromosomes, which is called an m-pair by analogy with that of Heteroptera. However, the m-chromosomes of the Odonata differ in that they exhibit regular behavior, form a chiasmatic bivalent, and differ only in size and occasionally in negative heteropycnosis [14]. The size of the smallest pair relative to the size of the immediately larger pair varies greatly from species to species, leading some authors to consider the former as the m-pair, even though the difference is small [31]. To avoid misinterpretation, Mola [14] suggested that only chromosomes smaller than or equal to half of the immediately larger pair should be defined as an m-pair.

The family Aeschnidae includes 405 species, of which only 58 have been analyzed cytologically [2,32]. The neo-sex chromosome system is common in this family and occurs in ten species from five genera: *Aeshna caerulea* (Ström, 1783), *A. grandis* (Linnaeus, 1758), *A. juncea* (Linnaeus, 1758), *A. serrata* (Hagen, 1856), *A. viridis* (Eversmann, 1836), *Anax ephippiger* (Burmeister, 1839), *Caliaeschna microstigma* (Schneider, 1845), *Gynacantha interioris* (Williamson, 1923), *R. bonariensis* (Rambur, 1842), and *R. planaltica* (Calvert, 1845) [32].

The genus *Rhionaeschna* (Förster, 1909) includes 42 species which are found in the Americas from southern Argentina to southern Canada. Most species occur in the Neotropics, with only three species having been described north of this region, while they are completely absent in the Amazon basin [2,33]. Seven species have been analyzed cytogenetically (Table 1) (all cited as *Aeshna*) [12,34,35,36].

Fluorescence in situ hybridization (FISH) using an 18S rDNA probe was performed in only 17 species of Anisoptera and Zygoptera [37,38,39]. Kuznetsova and colleagues [39] reported that the major rDNA clusters are located on one of the largest autosome pairs in Anisoptera and on the m-chromosomes in Zygoptera and considered the former to be the ancestral location for Odonata.

The objectives of the present study were to analyze the distribution of the nucleolar organizer region (NOR) using FISH with ribosomal DNA (rDNA) probes in *Rhionaeschna bonariensis*, *R. planaltica*, and *Aeshna cyanea* and to clarify the origin of the neo-XY chromosome determination system in *R. planaltica.* In addition, genomic in situ hybridization (GISH) was performed for the first time in Odonata, allowing accurate differentiation of sex chromosomes in *R. bonariensis*.

## 2. Materials and Methods

### 2.1. Insects

Adult males of *Rhionaeschna bonariensis* and *R. planaltica* were collected in the Martín García Island Provincial Nature Reserve, Buenos Aires Province, Argentina (34°11′15′′ S 58°16′52′′ W), and *Aeshna cyanea* was collected in České Budějovice, South Bohemia, Czech Republic (48°58′29″ N 14°28′29″ E).

### 2.2. Chromosome Preparations

For the FISH and GISH techniques, spread chromosome preparations were made from the testes of adult males. Gonads were dissected in a physiological solution, swollen in a hypotonic solution (0.075 M KCl) for 10 min, and then fixed in freshly prepared Carnoy’s fixative (ethanol, chloroform, acetic acid; 6:3:1) for 15–30 min. Cells were dissociated with tungsten needles in a drop of 60% acetic acid and spread on the slide with a heating plate at 45 °C, as described previously [40]. Then the preparations were dehydrated in an ethanol series (70, 80, and 96%, 30 s each) and stored at −20 °C until further use.

### 2.3. Fluorescent Banding

Fluorescent staining with GC-specific CMA_3_ (chromomycin A_3_) and AT-specific DAPI (4’-6-diamidino-2-phenylindole) was performed on unstained slides. A piece of the gonad was squashed in 45% acetic acid; the coverslip was then removed by the dry-ice method and the slide was air-dried. Sequential DAPI–CMA_3_ banding was performed according to the published protocol [41].

### 2.4. FISH with rDNA Probes

For *R. bonariensis* and *R. planaltica*, unlabeled 18S ribosomal DNA (rDNA) probes were generated by polymerase chain reaction (PCR) using universal arthropod primers: forward 5’-CCTGAGAAACGGCTACCACATC-3’ and reverse 5′ -GAGTCTCGTTCGTTATCGGA-3’ [42]. Total genomic DNA (gDNA) of *R. bonariensis* obtained by standard phenol-chloroform-isoamyl alcohol extraction was used as a template.

PCR was performed according to the previously described procedure [43]. The PCR product showed a single band of about 1000 bp. The band was purified using a QIAquick Gel Extraction Kit (Qiagen GmbH, Hilden, Germany) and used as a template for PCR amplification of 18S rDNA probes. Probes were labeled with biotin-14-dUTP by nick translation using a BioNick Labeling System (Invitrogen, Life Technologies Inc., San Diego, CA, USA).

FISH with a biotinylated 18S rDNA probe was performed according to the published procedure [44] with previously described modifications [43]. Briefly, denaturation of chromosomes was performed at 68 °C for 3.5 min in 70% deionized formamide in 2 × SSC. The probe cocktail for one slide (10 μL; 50% deionized formamide, 10% dextran sulfate in 2 × SSC) contained 45 ng of labeled probe and 25 μg of sonicated salmon sperm DNA. Hybridization was performed overnight. Hybridization signals were detected with Cy3-conjugated streptavidin and one round of amplification with biotinylated anti-streptavidin and Cy3-conjugated streptavidin. Preparations were counterstained with 0.5 μg/mL DAPI and mounted in DABCO-based antifade (Sigma-Aldrich, St. Louis, MO, USA).

For *A. cyanea*, FISH was performed with a 28S rDNA probe to detect NORs. The unlabeled 28S rDNA probe was obtained by PCR using gDNA from *Tityus argentinus* (Borelli, 1899) (Buthidae, Scorpiones) as a template, as described previously [45]. The probe was labeled by PCR with biotin-16-dUTP. FISH was performed as previously described [46] for indirect labeling. After overnight hybridization, the probe was detected with Cy3-conjugated streptavidin. Preparations were counterstained with 0.5 μg/mL DAPI and mounted in Vectashield Mounting Medium (Vector, Burlingame, CA, USA).

### 2.5. Genomic in Situ Hybridization (GISH)

Genomic DNA was isolated from male adults of *R. bonariensis* by standard phenol-chloroform –isoamyl alcohol extraction. Labeling was performed using a nick translation mix (Roche Diagnostics GmbH, Mannheim, Germany). DNA was labeled with Cy3-dCTP (red). GISH was essentially performed following the published procedure [47] for comparative genomic hybridization (CGH), except for the probe cocktail. For one slide, the cocktail contained 300 ng of labeled male gDNA of *R. bonariensis*, 3 μg of unlabeled sonicated male gDNA of *R. bonariensis*, and 25 μg of sonicated salmon sperm DNA in 83.5 μL of hybridization solution.

### 2.6. Microscopy and Image Processing

*Rhionaeschna* species preparations were observed in a Zeiss Axioplan 2 fluorescence microscope (Carl Zeiss, Jena, Germany). Black-and-white images were recorded with a cooled F-View CCD camera and captured with AnalySIS software, version 3.2 (Soft Imaging System GmbH, Münster, Germany). Preparations of *A. cyanea* were observed in a Leica DMLB fluorescence microscope equipped with a Leica DFC350 FX monochrome digital camera and Leica IM50 software, version 4.0 (Leica Microsystems Imaging Solutions Ltd., Cambridge, UK). Images of chromosomes were captured separately for each fluorescent dye and then pseudocolored (light blue for DAPI and red for Cy3) and superimposed using Adobe Photoshop, version 7.0 or Adobe Photoshop CS5.

## 3. Results

### 3.1. Localization of rDNA

Male meiosis of *Rhionaeschna bonariensis* and *R. planaltica* has already been described [12]. Briefly, in *R. bonariensis* (2n = 26, n = 12 + neo-XY), the neo-XY is the largest bivalent; it is heteromorphic at most meiotic stages and always has one terminal chiasma. The autosomal bivalents gradually decrease in size, except for the small m-bivalent. The neo-X chromosome arises from the terminal fusion of an autosome with the X chromosome, which is often observed in meiosis I with slightly separated chromatids [12].

In *R. bonariensis*, FISH with an 18S rDNA probe revealed a duplicated cluster of rDNA in pachytene, showing a pair of hybridization signals in the subterminal region of an autosomal bivalent (Figure 1a). In diplotene, it became clear that the hybridization signals corresponded to the largest autosomal bivalent. From this stage onwards, the signals were observed in a terminal position due to increasing chromosome condensation. Hybridization signals were always found at the opposite end of the chiasma, with the signal intensity varying in each telomeric region (Figure 1b–d).

*Rhionaeschna planaltica* (2n = 16, n = 7 + neo-XY) has a reduced diploid chromosome number and the chromosomes can be grouped into ten large chromosomes, five (four autosomes and neo-X) medium-sized chromosomes, and the smallest neo-Y chromosome. The sex chromosome bivalent is the smallest of the set and is distinctly heteromorphic because the X chromosome and one of the m-chromosomes are involved in its formation [12].

In *R. planaltica*, hybridization signals of the 18S rDNA probe indicate that the rDNA cluster is located in the subterminal region of a medium-sized chromosome during mitosis (Figure 2a,c). In pachytene, the small neo-XY bivalent has two regions of different sizes that are usually paired and a submedial loop with hybridization signals, indicating the absence of homology (Figure 2b,d,e). Occasionally, the smallest region is partially unpaired in late pachytene (Figure 2d). From diakinesis onwards, hybridization signals are observed in the region near the chiasma of the small sex chromosome bivalent (Figure 2f,g). DAPI–CMA_3_ staining at the pachytene stage showed a DAPI-dull/CMA_3_-bright band in the loop of the neo-XY pair (Appendix A).

*Aeshna cyanea* (2n = 27, n = 13 + X0) has a karyotype typical for the Aeshnidae. The chromosomes gradually decrease in size, except for the m-pair and the X chromosome, which is the second smallest of the complement [9]. In this species, which was used as an outgroup, the hybridization signals of the 18S rDNA probe indicate that the rDNA clusters are located in a secondary constriction of a large (although not the largest) autosomal pair (Figure 1e,f).

### 3.2. GISH Characterization of Neo-XY Sex Chromosomes in Rhionaeschna Bonariensis

GISH was used to determine whether the morphological differentiation of neo-sex chromosomes in *R. bonariensis* is also accompanied by differentiation at the molecular level, i.e., in DNA composition. In the GISH experiment, a labeled genomic probe from *R. bonariensis* males was hybridized to meiotic chromosome preparations of the same species with an excess of unlabeled genomic DNA from *R. bonariensis* males. At the pachytene stage, the male probe highlighted one of the members of the sex chromosome bivalent over most of its length (Figure 3a). Weak hybridization signals were also detected in some telomeric regions of the autosomal bivalents (Figure 3b). From diakinesis onwards, the smaller chromosome of the sex chromosome bivalent (neo-Y) showed strong hybridization signals (Figure 3c–e). All prometaphase and metaphase II stages had the heteromorphic neo-XY bivalent, which showed strong hybridization signals on the neo-Y chromosome and occasionally a small signal in the terminal region of the neo-X chromosome (Figure 3f). Distinct hybridization signals were observed in approximately half of the spermatids and were absent in the other half (Figure 3g,h).

Because the unlabeled competitor DNA and the probe were from the same species, the strong hybridization signals suggested the presence of accumulated repetitive DNA sequences in the neo-Y chromosome and in some telomeric regions. Despite the possibility that both the labeled and unlabeled DNA in the preparations could have bound to the neo-Y chromosome, these signals were clearly detectable, probably because the neo-Y chromosome contains multiple repetitive sequences that can bind to any of them. Conversely, the unlabeled competitor DNA may have masked the small amount of bound labeled DNA probe in the autosomes and the neo-X chromosome due to the lower repetitive content.

## 4. Discussion

Of the seven *Rhionaeschna* species studied, *R. californica* (Calvert, 1895), *R. confusa* (Rambur, 1842), and *R. peralta* (Ris, 1918) have the modal chromosome number of the family Aeshnidae, which is 2n = 27 (n = 13 + X0) in males. Compared to this modal karyotype, *R. bonariensis* shows a fusion of the X chromosome with an autosome. A reduction in chromosome number is observed in both *R. diffinis* (Rambur, 1842) and *R. intricada* (Martin, 1908) due to autosomal fusions and in *R. planaltica* due to fusions between autosomes and an autosome and the X chromosome (Table 1). The latter species has the karyotype with the lowest chromosome number in the genus *Rhionaeschna*, which could be the result of five fusions between non-homologous autosomes and from two pairs of the non-fused ancestral chromosomes. Furthermore, submedial insertion of the X chromosome into one of the m-chromosomes may have given rise to the neo-X chromosome, while the other m-chromosome became the neo-Y chromosome (see below).

A large reduction in chromosome number due to fusions has also been reported for other Aeshnidae species, either due to autosomal fusions as in *Anax guttatus* (Burmeister, 1839) (2n = 15, n = 7 + X0) or due to autosomal and X chromosome/autosome fusions as in *A. ephippiger* (Burmeister, 1839) (2n = 14, n = 6 + neo-XY) and *Caliaeschna microstigma* (Schneider, 1845) (2n = 16, n = 7 + neo-XY) [10,26,48,49].

Previous studies using FISH with an 18S rDNA probe show that NORs occur on an autosomal pair. In Zygoptera, NORs are generally present on the m-pair, but in *Ischnura elegans* they are located on a medium-large pair of autosomes [37,39]. In Anisoptera, NORs are found on a large autosomal pair [38,39], as in *R. bonariensis* and *Aeshna cyanea*, which were analyzed here. In contrast, in *R. planaltica*, the NOR is located on the neo-X chromosome (see below). Thus, the results of our study have improved our knowledge of the variation in the distribution of NORs in Odonata species.

Staining with fluorescent DNA-binding dyes with different specificities, such as DAPI and CMA_3_, allows characterization of heterochromatic regions in terms of AT- and GC-rich base composition, respectively. The association between CMA_3_-bright bands and NORs is common in insect groups with holokinetic chromosomes (e.g., Heteroptera, Homoptera, and Psocoptera), suggesting that rDNA is usually rich in GC repeats [19,20,50]. In Odonata, the CMA_3_-bright band associated with the NOR was reported for the largest autosomal pair in *Coryphaeschna perrensi* [51]. In the present work, a similar association was observed for the neo-X chromosome carrying NOR in *R. planaltica*.

Secondary constrictions are rarely found in holokinetic chromosomes. In Heteroptera, secondary constrictions have been described that are not associated with NORs [52]. However, the occurrence of an autosomal pair with a secondary constriction corresponding to the NOR has been reported in *Nezara viridula* and *Pachylis argentinus* (Heteroptera) [53,54]. This study shows that *Aeshna cyanea* has an NOR in a secondary constriction, which is a unique feature of Odonata.

### 4.1. Origin of the Neo-XY Sex Chromosomes in Rhionaeschna planaltica

In the case of *R. planaltica*, a detailed analysis of meiotic prophase and results of FISH led us to postulate the origin of neo-XY sex chromosomes. The submedial loop observed at the pachytene stage corresponds to the X chromosome inserted into the m-chromosome at about one-third of its length (Figure 4a–c). In mitosis, the NOR is located in the subterminal region of the neo-X chromosome, whereas in pachytene it is located in the loop of the X chromosome. In diakinesis, the NOR is located near the chiasma, which always occurs in the terminal autosomal segment of the neo-X chromosome (Figure 4d). Based on this observation, we propose that NOR is located in the subterminal region of the original X chromosome, which is connected to the smaller part of the original m-chromosome, and that the single chiasma in the bivalent occurs in this small segment. So far, the hypothesis that the neo-X chromosome originates as a result of transposition (insertion) has only been proposed for *R. planaltica* (this study) and the dragonfly *Orthemis ambinigra* [16].

The fact that the rDNA repeats are located on an autosomal pair in the other species studied suggests that during the evolution of the chromosome complement of *R. planaltica*, the rDNA was transposed from an autosome to the X chromosome by, for example, transposons or unequal crossing-over. In the next step, the other autosomal NOR was lost and the NOR on the X chromosome was retained, possibly by genetic drift or natural selection.

### 4.2. Characteristics of the Neo-XY Sex Chromosomes in Rhionaeschna bonariensis

In Odonata, characterization of the heterochromatin of neo-XY sex chromosomes has been performed by C-banding in only three *Aeshna* species [17,18]. In *A. viridis*, the neo-XY has large asymmetric heterochromatic blocks, the original X chromosome part of the neo-X chromosome is completely heterochromatic, and the autosomal segments of the neo-X and neo-Y chromosomes lack C-bands [17]. In *A. grandis* and *A. juncea*, the original X chromosome part of the neo-X chromosome has heterochromatic blocks in terminal positions and the autosomal segments of neo-X and neo-Y chromosomes contain a large amount of heterochromatin [18]. In the three *Aeshna* species, the formation of the neo-XY/neo-XX system was accompanied by heterochromatinization of the autosomal region, which was partial in the neo-X chromosome and complete in the neo-Y chromosome [18]. Of the three species, only *A. grandis* has a heteromorphic sex chromosome bivalent visible using conventional staining [9,10]. The neo-XY system of *R. bonariensis* differs from that of the species studied by Perepelov & Bugrov [18] due to the absence of heterochromatin in the neo-X chromosome. In contrast, it resembles neo-XY systems in Orthoptera, which have heterochromatin in only the neo-Y chromosome [55,56,57,58].

Based on our GISH results, which highlighted the neo-Y chromosome at all meiotic stages, and in agreement with the results of sex chromosome studies in other insect groups [43,56,59,60,61,62,63], we can conclude that the neo-Y of *R. bonariensis* consists mainly of constitutive heterochromatin with a high content of repetitive DNA sequences. The restriction of recombination to a single terminal region may have facilitated the accumulation of repetitive sequences by, for example, transposons or unequal crossing-over [64,65,66]. This likely led to changes in chromatin structure and subsequent heterochromatinization of the neo-Y chromosome. In late pachytene, given the large amount of repetitive DNA in the neo-Y chromosome, pairing of the neo-XY chromosomes would be mainly heterologous and would involve a phenomenon known as synaptic adjustment [67,68,69]. In addition, the presence of a large amount of heterochromatin and the location of the chiasma may indicate a very ancient origin of this neo-sex chromosome system.

The phylogenetic relationships of *Rhionaeschna* allow the species to be grouped into four clusters. *R. intricata* belongs to one cluster, *R. planaltica* to the second cluster, and the species from *R. californica* to *R. diffinis* to the third cluster [33] (Figure 5). In the fourth cluster, there are no species with a known karyotype. According to these phylogenetic relationships, *R. planaltica* and *R. bonariensis* are not closely related. In addition, the origins of their neo-XY system differ, with the system evolving by transposition in the former (insertion of the X chromosome into an m-chromosome) and by terminal fusion of an autosome with the X chromosome in the latter. Thus, it can be concluded that the neo-sex chromosomes of these two species evolved independently. Similarly, it can be concluded that the reductions in the number of autosomes observed in *R. intricata* and *R. diffinis* have independent origins (Figure 5).

## 5. Conclusions

The study of meiosis and the results of FISH with an 18S rDNA probe in *R. planaltica* allowed us to postulate that neo-XY sex chromosomes originated by insertion of the ancestral X chromosome into an autosome. In *R. bonariensis*, the results of GISH, which highlighted the neo-Y chromosome at all meiotic stages, indicate that the neo-Y chromosome consists mainly of constitutive heterochromatin with a high content of repetitive DNA sequences and has a very ancient origin. Considering the phylogenetic relationship of *R. planaltica* and *R. bonariensis* and the different origins of their neo-XY system, it can be concluded that the neo-sex chromosomes of these two species evolved independently. The present study provides new information regarding the origin and evolution of neo-sex chromosome determination systems in Odonata and suggests novel types of chromosome rearrangements, transposition of NORs, and heterochromatin accumulation.

## Figures and Tables

**Figure 1 insects-13-01159-f001:**
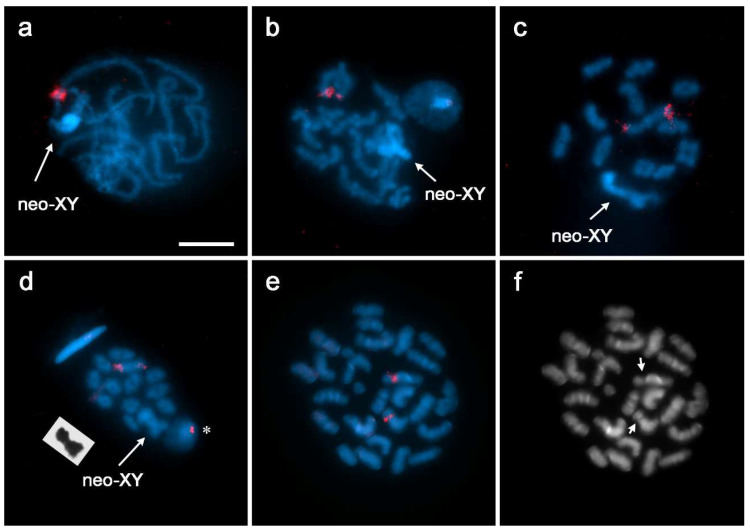
Localization of rDNA in *Rhionaeschna bonariensis* (n = 12 + neo-XY) (**a**–**d**) and *Aeshna cyanea* (2n = 27) (**e**,**f**) by FISH with 18S and 28S rDNA probes, respectively (red signals). (**a**) Pachytene; (**b**) early diakinesis; (**c**) diakinesis; (**d**) prometaphase I, inset: Giemsa-stained neo-XY bivalent from another cell, *: early spermatid with rDNA signal; (**e**,**f**) the same spermatogonial prometaphase: (**e**) composite FISH image; (**f**) DAPI staining; arrows point to secondary constrictions. Bar = 10 µm.

**Figure 2 insects-13-01159-f002:**
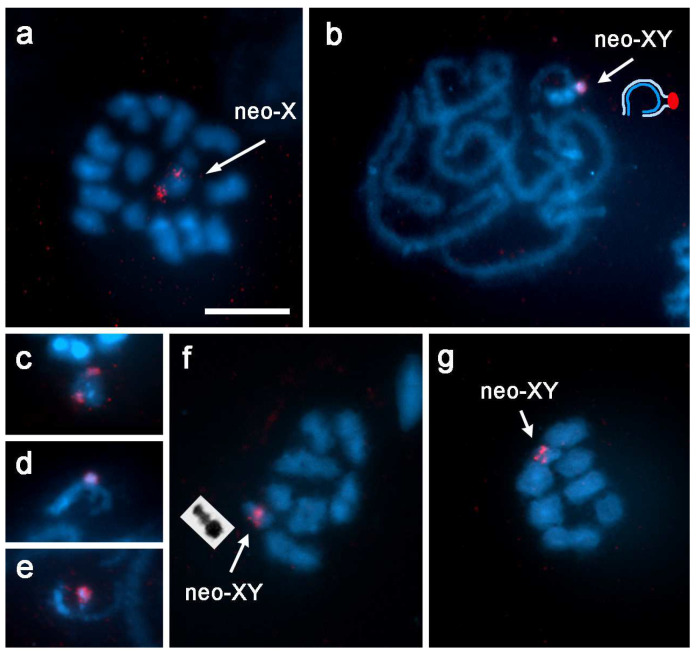
*Rhionaeschna planaltica* (2n = 16, n = 7 + neo-XY). Localization of rDNA by FISH with 18S rDNA probes (red signals) (**a**–**g**). Giemsa staining (inset in **f**). (**a**,**c**) Spermatogonial prometaphase; (**b**,**d**,**e**) pachytene; inset in (**b**): schematic drawing of the neo-XY bivalent (neo-X in white, neo-Y in blue, probe signals in red); (**f**) diakinesis, inset: neo-XY bivalent; (**g**) metaphase I; (**c**) shows a neo-X chromosome to illustrate the subterminal position of the signal; (**d**,**e**) show selected neo-XY bivalents from other cells. Bar = 10 µm.

**Figure 3 insects-13-01159-f003:**
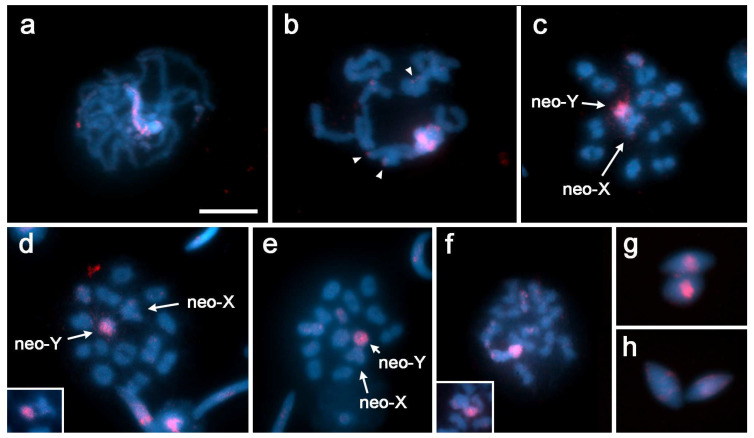
GISH in *Rhionaeschna bonariensis* (n = 12 + neo-XY). The male gDNA probe was labeled with Cy3-dCTP (red). (**a**) Pachytene; (**b**) diplotene; (**c**,**d**) diakinesis; (**e**) prometaphase I; (**f**) prophase II; (**g**,**h**) spermatids with (**g**) and without (**h**) hybridization signals. Arrowheads in (**b**) indicate weak hybridization signals in some telomeric regions of autosomal bivalents. Insets in (**d**,**f**) show selected neo-XY bivalents from other cells. Bar = 10 µm.

**Figure 4 insects-13-01159-f004:**
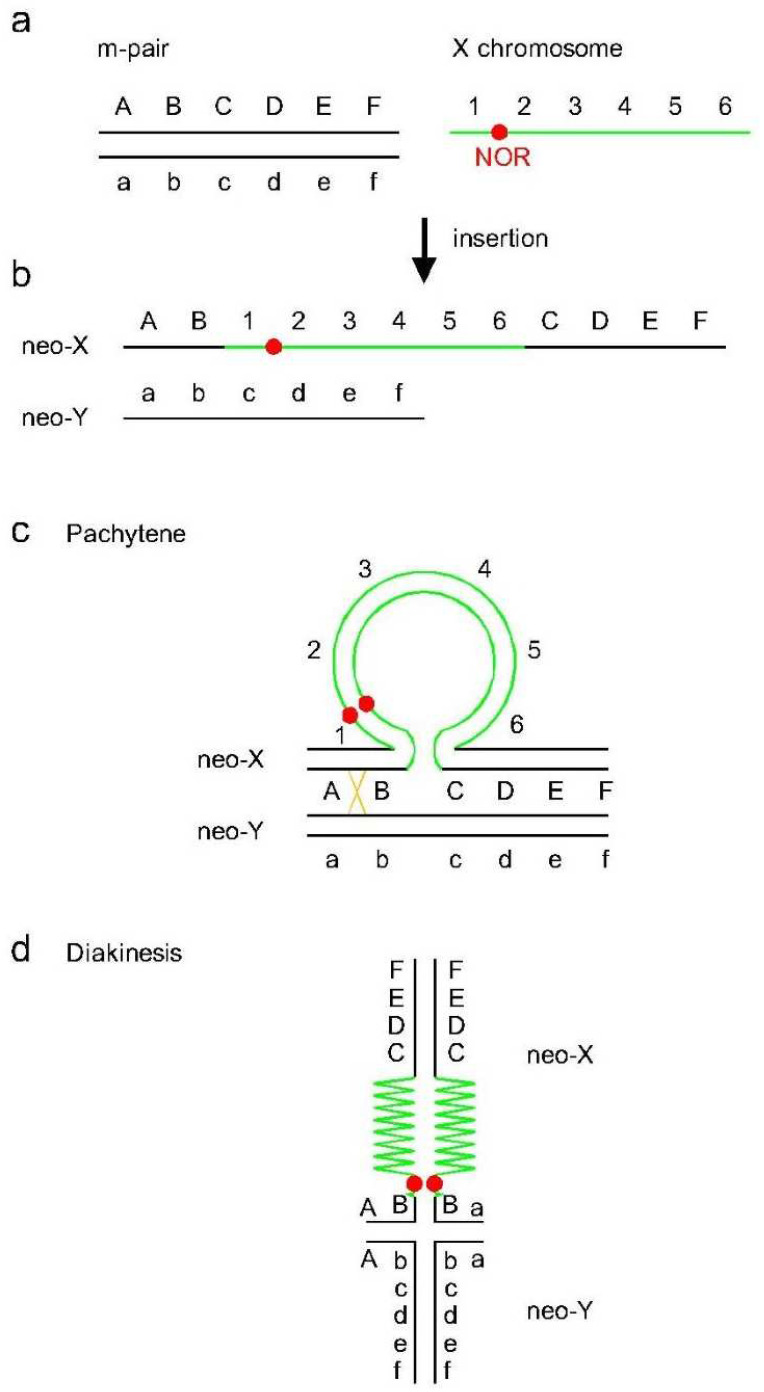
Schematic interpretation of chromosomal rearrangements in the evolution of neo-X and neo-Y chromosomes and meiotic configurations in *Rhionaeschna planaltica*. (**a**) Autosomal m-chromosome pair (black) and single X chromosome (green) with subterminal rDNA cluster, i.e., NOR (red). (**b**) Insertion of the X chromosome into one-third of one m-chromosome (neo-X). The NOR is located in the region of the X chromosome that was connected to the smaller part of the m-chromosome. (**c**) Pachytene pairing of the neo-XY bivalent with a crossing-over (orange) in the smaller pairing segment. (**d**) Heteromorphic sex chromosome bivalent in diakinesis with NOR near the chiasma.

**Figure 5 insects-13-01159-f005:**
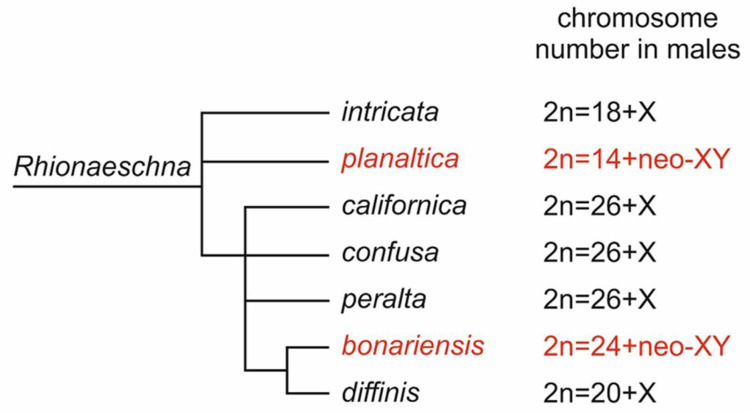
Simplified phylogenetic tree of *Rhionaeschna* species with known diploid chromosome number and sex chromosome constitution in males constructed according to the cladogram published in a previous study [33]. Species examined in this study are shown in red.

**Table 1 insects-13-01159-t001:** Summary of cytogenetic data in males of *Rhionaeschna* species.

Species	2n	n	Locality	References
*R. bonariensis* (Rambur, 1842)	26	12 + neo-XY	Argentina, Uruguay	[12]
*R. californica* (Calvert, 1895)	27	13 + X0	Canada	[35]
*R. confusa* (Rambur, 1842)	27	13 + X0	Argentina, Uruguay	[12]
*R. diffinis* (Rambur, 1842)	21	10 + X0	Bolivia	[34]
*R. intricata* (Martin, 1908)	19	9 + X0	Bolivia	[34]
*R. peralta* (Ris, 1918)	27	13 + X0	Bolivia	[34]
*R. planaltica* (Calvert, 1845)	16	7 + neo-XY	Argentina	[12]

## Data Availability

Details of the data presented in this study are available on request from the corresponding authors.

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
