# Peer review of "On the Origin of Neo-Sex Chromosomes in the Neotropical Dragonflies Rhionaeschna bonariensis and R. planaltica (Aeshnidae, Odonata)"

_insects, 2022, doi:10.3390/insects13121159_

Round 1

Reviewer 1 Report

The paper aims to analyze the distribution of NOR regions and heterochromatin in three species Rhionaeschna bonariensis, R. planaltica and Aeshna cyanea, through FISH and Genomic in situ hybridization. GISH was performed for the first time in Odonata. The results also allowed the authors to determine the origin of the neo -sex chromosomes in the two Rhioaeschna species and that they both have different origins for the neo-XY system.

The manuscript is well written and clear and presented in a structured manner. The references are relevant and most of them are recent. The manuscript only needs minor revision for publication.

I only have a few comments that I have also posted in the pdf.

In the Introduction, line 91- The authors should give some references for the different opinions about m-pair size and identification.

Material and methods – the subchapter title “FISH with 18S rDNA Probes” should also include 28S rDNA probes or “FISH with rDNA Probes “

Figure 5 caption is misleading. It should be stipulated that it is a simplified phylogenetic tree made after von Ellennrieder, 2003.

Reviewer 2 Report

The authors focused on three Aeshnidae species to understand the evolution of neo-sex chromosomes. They analyzed NOR by FISH with rDNA probes and conducted GISH. Abstract and simple summary are easy to understand and very interesting to me. However, it is very difficult for me to interpret the result of the figures and understand the context maybe partly due to the lack of explanations. Moreover, the discussion section seems to include overclaims. Especially, Fig.4 is a very beautiful hypothesis, but I cannot derive this hypothesis from the result figures.

In this manuscript, they revealed the sex-chromosome system in two species, but both results are difficult for me to understand. I’m sorry to say that I’m not a super-expert in cytology, but I think readers of this Insects journal include entomological researchers in different areas. So, please check my questions below and please add the explanation to avoid readers’ misunderstanding if needed.

<Results>

1. L190-195, are these sentences from the previous study [11]? If these sentences are derived from Fig.1, I cannot understand why you concluded as you mentioned.

2. L196, I cannot figure out why arrow regions (Fig. 1) can be identified as neo-XY, and why 18S rDNA signals are on the autosomal bivalent.

3. L217, it is very difficult for me to understand “a submedial loop with hybridization signals” from Fig. 2b.

4. L221, as mentioned above, I cannot figure out why arrow regions (Fig. 2b, f, g) can be identified as neo-XY.

5. L241, from Fig.3b, I cannot understand “Hybridization signals were also detected in some telomeric regions of the autosomal bivalents”. Also, which are the hybridization signals in some telomeric regions of autosomal bivalents in Fig.3b? Please add arrowheads in Fig.3b.

6. L221, as mentioned above, I cannot figure out why arrow regions (Fig. 3c,d,e) can be identified as neo-XY.

7. L258, is it true? It seems to me that neo-Y specific red signals are derived from Y chromosome sequence specificity, not from repetitive sequences.

<Discussions>

1. L264-280, what is the important point you want to say here for the discussion derived from your results?

2. L288-294, if you want to add this paragraph, you should add the data with DAPI/CMA3 data (I think “data not shown” should be avoided).

3. L304-, why you can say “The submedial loop observed at the pachytene stage corresponds to the X chromosome inserted into the m-chromosome at about one-third of its length (Figure 4a–c)”. And also, from your figures, I cannot understand why m chromosome is important.

Minor points are below:

Line65: You should add brief explanations of “holokinetic chromosomes” and “post-reductional meiosis” with appropriate citations.

Line67: about 95% of species -> about 95% of Odonata species (?)

Line88-91: You should add appropriate citations for m-pair chromosomes.

Line109-110: You should add an appropriate citation for this sentence.

Line110: [30] -> Different paper (?) Maybe “Chromosomal analysis of eight species of dragonflies (Anisoptera) and damselflies (Zygoptera) using conventional cytogenetics and fluorescence in situ hybridization: Insights into the karyotype evolution of the ancient insect order Odonata”

Line 140: 18S rDNA -> rDNA (Because you used 28S rDNA for A. cyanea)

Reviewer 3 Report

This paper provides the interesting evidence that two dragonflies of the genus Rhinoaeschna have independent neo-XY chromosome origins. My comments are as follows:

>line 37, Abstract:

Please specify the full name of NOR first.

>line 84, Introduction

Hemianax -> Anax

>line 99

nine species from four genera -> ten species from five genera

Please add Anax ephippiger.

>line 223 data not shown

Please show the results.

>line 237

The authors should first briefly explain the purpose of doing GISH.

Reviewer 4 Report

The authors have studied the origin of neo-sex chromosomes by examining the distribution of NOR using FISH with rDNA probes in three Odonata species, Rhinoaeschna bonariensis, R. planaltica, and Aeshna cyanea, and applied genomic in situ hybridization (GISH) for the first time in Odonata. The authors have shown that the neoX has arisen by a terminal fusion of the original X with an autosome in R. bonariensis and A. cyanea, while in R. planaltica, the neoX has originated by interstitial insertion of the original X in the smallest autosome. In addition, the authors showed that the neoY chromosome in R. bonariensis consists of multiple repetitive sequences. The methods have been applied skilfully and the results are documented well. Figure 4 is highly informative and helps the reader to understand X chromosome insertion and how it can be observed in pachytene and diakinesis stage.  I have only one remark. In figure legend for figure 1, it is said that fig. 1 f represents Giemsa staining. However, it appears that the figure shows a prometaphase stained with CMA3.

There is, however, one serious problem in the manuscript. In introduction, the authors write “Odonata are characterized by holokinetic chromosomes and post-reductional meiosis.” without any references. This kind of expression is like giving a trivial fact, which it is not. Looking for a paper where this kind of conclusion has been presented, an article by the first author was found (Mola, Hereditas 122:47-55, 1995). In the paper it is suggested that if the longitudinal orientation of the bivalent is parallel to the spindle axis it is a sign of pre-reductional meiosis, but if it is parallel to the equatorial plane, it is a sign of post-reduction. However, the type of meiosis cannot be inferred from the morphology of the bivalent (Nokkala and Nokkala, Heredity 78: 561-566, 1997; Viera et al. In: Meiosis, vol 5: 137-156, 2009). In fact, meiosis in Odonata is pre-reductional (Nokkala et al., Hereditas 136: 7-12, 2002).

It is important to notice that the mechanism in starting anaphase I and the segregation of half bivalents in the first meiotic division in the pre-reductional meiosis includes sister chromatin cohesion release from those chromosome regions oriented along the equatorial plane. The extent of these regions depends on the location of crossing over in the homologous chromosomes. This division of chromosome regions must not be confused with the equational division of those half bivalents in the second meiotic division. 

My suggestion is to either replace post-reductional meiosis with pre-reductional meiosis or not to refer to the type of meiosis at all as it is not crucial in the connection of the present study.

Round 2

Reviewer 2 Report

Authors answered my question in brief, so I can understand what I was confused. Improved.